# OpenReview forum: "Towards Understanding Robustness and Generalization in World Models"
_ICLR.cc/2025/Conference — ICLR 2025 Conference Withdrawn Submission_

### Official Review · Reviewer_37oG · 2024-10-22

**Soundness:** 3
**Presentation:** 1
**Contribution:** 2
**Rating:** 3
**Confidence:** 3

**Summary:**

This paper investigates how regularization can give rise to better generalization in world models, particularly how Latent Representation Errors can be an important source for robustness in Dreamer-style world models. Theoretical results are derived and authors propose a Jacobian regularization scheme for improving robustness. Their approach is tested on two Mujoco environments with noise perturbations and augmented versions of image observations.

**Strengths:**

* Interesting research question: can world model inaccuracies improve generalization.
* Jacobian regularization of Dreamer-style models is, to my knowledge, novel.
* Proposed method improves robustness.

**Weaknesses:**

While I appreciate that the paper's main contribution is theoretical, I think the paper is written in a way that makes it unnecessarily difficult to understand. The mathematical treatment is very thorough, but the readability of the paper suffers a lot as a consequence. Moreover, while a lot of attention is put into describing mathematical details that I don’t believe need to be in the main text, some very important concepts like Latent Representation Errors (LRE) and zero/non-zero drift and Jacobian regularization are discussed very briefly, and very late in the paper. In fact, LRE is mentioned many times in the introduction without ever being explained. The authors need to explain precisely what LREs are, how they differ from noise injection, and why LREs are an interesting source of robustness early on in the paper, and in a way that is accessible to most ML practitioners working on World Models. The same holds for drift and Jacobian regularization. A paragraph summarizing in natural language what these things are, and how they relate to each other would improve readability immensely. The subsequent mathematical treatment will be a lot easier to understand as a consequence.

Secondly, while the results look promising, they are not very extensive. Does the regularization give performance gains in other mujoco environments too? How does the performance compare to other ways of regularizing the dynamics model, like Repo [1] or RPC [2], or just L2 regularization on dynamics weights?

Due to the inadequate exposition of the paper, and the limited experimental evaluation, I cannot recommend it for acceptance in the current form. If the readability of the paper is significantly improved, and experiments in more DMC environments are added, and more baselines (like Repo and RPC) comparisons are made, I am willing to increase my score.

[1] Repo: Resilient model-based reinforcement learning by regularizing posterior predictability, Zhu et al., NeurIPS 2023

[2] Robust Predictable Control, Eysenbach et al., NeurIPS 2021

**Questions:**

* Did you evaluate your method on other Mujoco environments?
* Did you compare your method to other regularization methods?

---

> ### Author Response · Authors · 2024-11-20
>
> Thank you for your feedback and suggestions!
>
> A1: _Readability of the paper_
>
> We respectfully disagree with the statement. In fact, we made deliberate efforts to introduce and contextualize these concepts early in the text. Specifically, we describe LREs as errors incurred by latent encoders (in Line 71), and we explain their influence on robustness and generalization (Lines 48-49: "... investigate how latent representation errors can enhance robustness against perturbations, which in turn often improves generalization"). We also outline how LREs differ from conventional noise injection schemes (Lines 71-87), emphasizing their inherent nature and the challenges of error propagation during rollouts. Furthermore, the introduction establishes the relevance of zero/non-zero drift cases and the role of Jacobian regularization in mitigating the adverse effects of biased latent errors (Lines 99-103: For the case where latent representation errors exhibit non-zero drifts, … propose to add Jacobian regularization to mitigate these effects.").
>
> Nonetheless, we are open to further clarifying these points to enhance readability.
>
> A2: _“Secondly, while the results look promising, they are not very extensive. Does the regularization give performance gains in other mujoco environments too? How does the performance compare to other ways of regularizing the dynamics model, like Repo [1] or RPC [2], or just L2 regularization on dynamics weights?”_
>
> We emphasize that our work focuses on deepening the theoretical understanding of the generalization capabilities of world models—an area previously observed in empirical studies but lacking systematic theoretical analysis. Our experiments are specifically designed to support and validate the theoretical insights presented. For example, our "Robustness against Encoder Errors" experiment directly corroborates Corollary 3.8, which demonstrates how the model’s Jacobian norm can control the impact of errors on the loss $\mathcal{L}$.  We acknowledge that comparing our method with other regularization techniques, such as L2 regularization, Repo, or RPC, would be valuable for a future work with a broader empirical scope. We want to respectfully emphasize that the primary goal of this work is to advance the theoretical understanding of latent representation error within the specific context of world models, rather than to conduct a broad empirical comparison.

---

> > ### Comment · Reviewer_37oG · 2024-11-22
> >
> > Thanks for the response. I do still believe that the paper would be significantly improved by clearly explaining what LREs are and what causes them. Do they arise from noise in the observations? Or do they arise due to mismatches between dynamics predictions (e.g. priors) and observations? A simple explanation of what the authors mean by LRE in the introduction would improve the exposition so much - this is missing from the paper currently.
> >
> > Secondly, even though the main contribution is theoretical, it's hard to assess how effective the Jacobian regularization is when no other baseline regularization method is presented. Showing that it works better than L2 norm minimization of encoder weights is the least that can be done here.
> >
> > I have chosen to maintain my score for now.

---

> > > ### Author Response · Authors · 2024-11-22
> > > **latent representation error and Jacobian Regularization**
> > >
> > > We  appreciate your prompt response to our rebuttal, and thank you for this further opportunity that allows us to improve the clarity of our work.
> > >
> > > As is standard in representation learning, "latent representation error (LRE)" here is the mismatch between the model representation in latent space and the underlying model of the input data. Per your suggestion, we will add a statement in the Introduction.
> > >
> > > Thank you for your comment on Jacobian Regularization. We would like to clarify that one primary contribution  in this paper is the theoretical understanding of the impacts of non-zero drift error on instability in world models, which shed light on how the well-known Jacobian regularization technique can be employed to mitigate this issue. We did not claim Jacobian regularization as a novel contribution but rather as a theoretically grounded approach to address the challenges we have identified. To avoid any possible misinterpretation, we revised the draft to make this distinction more explicit in the introduction (on Line 102-104, Line 106-109, highlighted in red). Because we do not claim Jacobian regularization as our contribution,  we feel there is no need to add other baseline regularization method in this context.
> > >
> > > We hope this has addressed your concern.

---

### Official Review · Reviewer_hhzr · 2024-10-22

**Soundness:** 3
**Presentation:** 2
**Contribution:** 3
**Rating:** 5
**Confidence:** 3

**Summary:**

This paper theoretically analyzes the robustness and generalization of world models, particularly by formulating latent dynamics models as stochastic differential equations with latent representation errors. The authors first demonstrate that these errors can be made sufficiently small using appropriate CNN encoders and decoders. They then show that these errors introduce additional terms in the loss function, which act as implicit regularization in the zero-drift case but introduce unstable bias in non-zero-drift scenarios. To enhance robustness and generalization, the authors propose an explicit regularization on the Jacobian, which is empirically validated in Mujoco environments.

**Strengths:**

1. To my knowledge, there is currently no substantial theoretical analysis of latent dynamics models, despite the significant empirical success of models like Dreamer.
2. The theoretical analysis leads to the derivation of a regularization term that improves empirical performance.

**Weaknesses:**

I must preface my comments by stating that I am not an expert in theoretical analysis. As a researcher focused on empirical methods (such as Dreamer), my main concern is that the theoretical formulation presented in this paper may not align well with the empirical implementation of Dreamer. I have outlined my questions below. The Area Chair is free to disregard my review if they deem these points inappropriate, but I believe my perspective could be valuable in assessing this paper.

**Questions:**

1. What are the experimental settings in Table 1? I could not find any explanation or references in the Introduction.
2. There are various types of errors in latent dynamics models. In my opinion, the most crucial is the error in the transition models. However, this error is not addressed in equations (5-6). Could the authors provide insights into why they focus solely on representation errors?
3. What is the ground truth probability measure $P$ introduced in Line 232? In Dreamer, latent representations are learned using a per-sample reconstruction loss of states and a prediction loss. How do these implementation losses relate to the two minimizations of distribution distance mentioned in Line 238? Additionally, how does the $W_1$ distance relate to the scale of errors $\varepsilon$?
4. How is Corollary 4.2 regarding Q functions in Section 4 defined and derived? There do not appear to be any reward variables in the SDE formulation.
5. How is the Jacobian regularization term in equation (20) implemented in the experiments? I could not find implementation details in the Appendix.

I am open to discussing these questions with the authors and am willing to increase my rating if my questions are well responsed.

---

> ### Author Response · Authors · 2024-11-20
>
> Thank you for your feedback and questions!
>
> A1: _“What are the experimental settings in Table 1? … “_
>
> We  clarify that Table 1 is extracted from experimental studies presented   in the experiment section in Section 5 with details presented in the appendix. For the batch-size versus robustness experiment in Table 1, the considered perturbation methods are the same as those in the experiment on Jacobian regularization with perturbed states (Table 2 and Appendix D.2).
>
> A2: _“There are various types of errors in latent dynamics models. In my opinion, the most crucial is the error in the transition models. However, this error is not addressed in equations (5-6). Could the authors provide insights into why they focus solely on representation errors?”_
>
> Thank you for highlighting this perspective. In this work, we focus on latent representation errors because they are unique to world models that learn a low-dimensional latent state representation, unlike conventional model-based RL approaches. We also want to emphasize that the most significant impact—whether from latent representation errors or transition model errors—often arises from error accumulation over time rather than from the errors themselves at each individual step. During both training and predictive rollouts, even small errors can compound significantly across the trajectory of the dynamics model's predictions.
>
> Our main results can, in fact, be extended to address inaccuracies from the transition model's predictions. Specifically, to handle additional errors from the transition model, one could generalize our framework by incorporating these errors into Theorem 3.7 and Corollary 4.2. For example, one could incorporate error terms $\sigma$ as $(\sigma_{\text{enc}}, 0, \sigma_{\text{pred}}, 0)$ and $\bar{\sigma}$ as $(\bar{\sigma_{\text{enc}}}, 0, \bar{\sigma_{\text{pred}}}, 0)$. We are happy to elaborate further if needed.
>
> A3: _“What is the ground truth probability measure P introduced in Line 232?”_
>
> We assume that P is a probability measure supported on Z with its Radon-Nikodym derivative p w.r.t $\mu_\mathcal{Z}$. In practice, P is typically an easy-to-sample distribution, such as a uniform distribution, which is determined by the specific choice of encoder-decoder architecture (Lines 232-235).
>
> _“In Dreamer, latent representations are learned using a per-sample reconstruction loss of states and a prediction loss. How do these implementation losses relate to the two minimizations of distribution distance mentioned in Line 238?”_
>
> As discussed in Lines 242-246,  we use Wasserstein-1 distance to measure the discrepancy between probability measures. Specifically, for the Dreamer-type loss function that uses KL-divergence, we note that the squared W1 distance between two probability measures can be upper bounded by their KL-divergence up to a constant (Gibbs and Su, 2002). This implies that minimizing KL-divergence in the model could reasonably be expected to also reduce the W1 distance.
>
> _“Additionally, how does the distance relate to the scale of errors?”_
>
> The Wasserstein-1 distance measures the discrepancy between the target distribution and the represented distribution, which indicates the scale of approximation error in latent representation.
>
>
> A4: _“How is Corollary 4.2 regarding Q functions in Section 4 defined and derived? There do not appear to be any reward variables in the SDE formulation.”_
>
> Corollary 4.2 pertains to the currently learned Q function in policy learning phase in world models, which does not need to be explicitly defined in terms of reward variables. The corollary quantifies the impact of latent representation errors on this Q function during the inference phase of the latent dynamics model. Specifically, it shows that latent representation errors can implicitly encourage exploration of unseen states by introducing stochastic perturbations into the value function.
>
> A5: _“How is the Jacobian regularization term in equation (20) implemented in the experiments? I could not find implementation details in the Appendix.”_
>
> As mentioned in Lines 1574-1575, we use a computationally efficient approximation of the Jacobian norm for the sequence model. Specifically, we follow the method described in Hoffman et al. (2019), employing a single projection to approximate the Jacobian regularization term.

---

> > ### Comment · Reviewer_hhzr · 2024-11-25
> >
> > I appreciate the detailed response provided by the authors and apologize that I still need more clarification on A3:
> >
> > - Can the author provide a more intuitive description of the latent representation learning formulation? In Dreamer, for each $s_t\sim Q$, we need to minimize its reconstruction error and prediction error, which can be approximated written as $\min ||s_t - dec(enc(s_t))|| + KL(transit(enc(s_t)), enc(s_{t+1}))$. How does this correspond to Line 238, which seems to minimize the distance of two distributions $enc\sharp Q$ and $P$?
> > - And how the latent representation error $\varepsilon$ in Eq (5) can be "understood as the “difference" of pushforward measure by the encoder/decoder and target measure"? What is the exact relationship between $W_1$ distance and $\varepsilon$?

---

> > > ### Author Response · Authors · 2024-11-25
> > >
> > > Thank you for your thoughtful and detailed questions! We are happy to provide further clarification on both points.
> > >
> > > _Q1_: Our work focuses on latent representation learning, which we abstractly formulate as the process of encoding the state distribution $Q$ into a lower-dimensional latent space represented by $P$, using the encoder-decoder structure. Intuitively, this can be understood as the encoder-decoder's capacity to approximate $Q$ through $P$, minimizing discrepancies between the two distributions. Specifically, the reconstruction error—an error component describing latent representation—can be decomposed to $W1(g_\text{enc \\#} Q,P)$ and $W1(Q, g_\text{dec \\#} P)$. This reflects the encoder-decoder's ability to align $Q$ with $P$ through their respective mappings. For a visual illustration of this process, please see Figure 2 in Appendix A.
> > >
> > > In addition, specifically for KL-type reconstruction loss, the squared W1 distance can be upper bounded by their KL-divergence up to a constant (Gibbs and Su, 2002). This implies that minimizing KL-divergence in the model could reasonably be expected to also reduce the W1 distance.
> > >
> > > _Q2_: In Equation (5), the latent representation error is represented as $\epsilon\sigma(h_t, s_t))dt + \epsilon \bar{\sigma}(h_t, s_t)dB_t$, where $\epsilon$ quantifies the magnitude of the latent representation error. The W1 distance, as used on Line 238, is a commonly adopted metric for comparing distributions. A larger W1​ distance between $g_\text{enc \\#} Q$ and $P$ indicates a greater misalignment between the encoded state and the target distribution, corresponding to a larger $\epsilon$.
> > >
> > > We hope this clarifies your questions.

---

> > > > ### Comment · Reviewer_hhzr · 2024-11-27
> > > >
> > > > Thank you for your detailed response. However, I must apologize as the latest reply appears to largely reiterate the initial rebuttal without addressing some of my confusion. I remain unclear about the formulation of latent representation learning, which seems to align with similar concerns raised by Reviewer 37oG. Could the authors please clarify the following?
> > > >
> > > > **Q1 (continued):** Consider a scenario where the state space consists of only two states, with $ Q = \frac{1}{2}\delta(s_1) + \frac{1}{2}\delta(s_2) $ representing a uniform distribution. These states are encoded by a deterministic encoder such that $ \text{enc}(s_1) = z_1 $ and $ \text{enc}(s_2) = z_2 $, and decoded by a deterministic decoder such that $\text{dec}(z_1) = s_2 $ and $ \text{dec}(z_2) = s_1 $. In this case, $ P = \frac{1}{2}\delta(z_1) + \frac{1}{2}\delta(z_2) $, and both $ W_1(\text{enc}(Q), P) $ and $W_2(Q, \text{dec}(P)) $ reach their minimum value of 0. However, proper representation learning is not achieved because the decoder reconstructs incorrect states compared to the input states.

---

> > > > > ### Author Response · Authors · 2024-11-27
> > > > >
> > > > > We appreciate the reviewer’s thoughtful example, and address the specific comments as follows.
> > > > >
> > > > > _Clarification of the formulation for latent representation learning_:
> > > > > In our framework (outlined in Line 236 of the manuscript), we focus on encoder-decoder pairs that satisfy the W1 objective, forming a set of solutions for encoder-decoder maps. As said in our earlier response, our main objective is to quantify the encoder-decoder's capacity to approximate Q through P, minimizing discrepancies between the two distributions. There could be many solutions in such a  set. Worth noting is that not every candidate solution would yield ideal representation learning (as is the case in the example by the reviewer.)  Our approximation theorem (Theorem 3.6) then demonstrates that CNN-type maps can approximate these solutions with a small error. Importantly, we do not explicitly select a specific solution from this set but rather show the feasibility of approximating them.
> > > > >
> > > > > _Motivation for using the W1 objective_:
> > > > > We appreciate the thoughtful example provided by the reviewer and would like to highlight an important distinction in our setting: In the general settings we consider, the state space is not diffeomorphic to the latent space, simply because the state space is likely of high dimension and this is an underlying motivation why latent representation is often used to map the high-dim inputs to low-dim latent space. This makes an exact identity mapping via the encoder-decoder pair impossible—unlike the example provided. This distinction motivates our focus on reconstructing probability measures (as described by the W1 objective, with the existence of such reconstructions established in Proposition A.7) rather than perfect state-by-state reconstruction which may not exist.
> > > > >
> > > > > We hope we have addressed your concerns adequately.

---

> > > > > > ### Comment · Reviewer_hhzr · 2024-11-29
> > > > > >
> > > > > > I appreciate the author's efforts to address my confusion. However, after many rounds of discussion, I still find that the formulation's presentation is not clear, which led to my final decision of leaning rejection.

---

> > > > > > > ### Author Response · Authors · 2024-12-03
> > > > > > >
> > > > > > > Thank you for the comment and for engaging deeply with our work.
> > > > > > >
> > > > > > > We would like to clarify  further an important aspect regarding our formulation: As noted in Line 226, our case is when $d_M << d_S$, where the latent state space $Z$ (defined in Line 232) has a much lower dimension than the state space (as stated in Line 179). Furthermore, Z is a closed ball in $\mathbb{R}^{d_M}$. This dimensionality disparity makes an identity map infeasible in the example provided. More broadly, this difference arises from the core motivation behind using latent representations: to map high-dimensional inputs to a lower-dimensional latent space.
> > > > > > >
> > > > > > > Clearly, perfect state-by-state identity reconstruction in your example does not satisfy the assumptions. Please note that  we focus on reconstructing probability measures, as described by the W1 objective, with the existence of such reconstructions established in Proposition A.7.
> > > > > > >
> > > > > > > Again, thank you for reviewing our work.

---

### Official Review · Reviewer_Eonw · 2024-11-03

**Soundness:** 2
**Presentation:** 1
**Contribution:** 2
**Rating:** 3
**Confidence:** 4

**Summary:**

This paper studies robustness and generalisation in model-based RL agents with world models (e.g. DreamerV2/V3) by analysing the effects of latent representation errors. The authors develop a stochastic differential equation (SDE) formulation as continuous generalisation of discrete-time latent transitions in world models, and characterise two types of latent representation errors: zero-drift and non-zero-drift. Their key findings suggest that modest zero-drift errors can actually act as implicit regularization and improve robustness, while non-zero-drift errors can be harmful. They propose using Jacobian regularisation to mitigate negative effects from non-zero-drift errors. The authors empirically evaluate the performance of the proposed regularisation methods on selected MuJoCo tasks.

**Strengths:**

- Novel theoretical perspective on applying the continuous-time formulation of discrete transition dynamics to latent transitions in world models.
- The proposed Jacobian regualrisation is simple and seems to lead to performance increase empirically.

**Weaknesses:**

- The presentation is quite dense and difficult to follow, with repetitive statements and inadequate motivation for why this analysis approach is needed.
- The assumption of requiring C3 functions with bounded Lipschitz derivatives for theoretical analysis can be very strong and potentially unrealistic in practice.
- Empirical evaluation is limited to only Walker and Quadruped environments, leading to imcomprehensive evaluations.
- The hyperparameter sensitivity for the regularization weight, $\lambda$, is expected to significantly affect the resulting model performance, but its sensitivity is not discussed (either theoretically or empirically) in the manuscript.
- There is a lack of comparison with existing approaches on addressing robustness and generalisability in world models.

**Questions:**

See Weaknesses.

---

> ### Author Response · Authors · 2024-11-20
>
> Thank you for your feedback!
>
> A1: _“The presentation is quite dense and difficult to follow, with repetitive statements and inadequate motivation for why this analysis approach is needed.”_
>
> We respectfully disagree with this statement. (As  a matter of fact, all three NeurIPS reviewers think this presentation is easy to follow.) We intentionally structured the introduction to provide a comprehensive and layered explanation of our approach, aiming to make our contributions clear and accessible to a broad audience. Specifically, we motivate our analysis approach by emphasizing the practical importance of understanding the robustness and generalization capabilities of world models (Lines 31-45) and explaining how our work addresses open questions in this area (Lines 47-70). Additionally, we succinctly summarize our main contributions in bullet points (Lines 94-111) to ensure clarity and to highlight the novelty and relevance of our work.
>
> That said, we greatly value the reviewer's perspective and try best to improve the clarity of our presentation. We would sincerely appreciate more specific suggestions or examples of sections that feel overly dense or repetitive, or guidance on how the motivation could be more effectively articulated.
>
> A2: _“The assumption of requiring C3 functions with bounded Lipschitz derivatives for theoretical analysis can be very strong and potentially unrealistic in practice.”_
>
> We respectfully note that assuming $\mathcal{C}^3$ with bounded Lipschitz derivatives is  a common practice in theoretical analyses of general neural networks (see, for example, Assumption A in [1]). This assumption encompasses widely used activation functions, such as tanh, which is frequently employed in RNNs for world models. While the assumption does exclude certain unbounded activation functions, those functions are prone to issues like exploding gradients and are therefore less commonly used in practical settings.
>
> A3: _“Empirical evaluation is limited to only Walker and Quadruped environments, leading to imcomprehensive evaluations.”_
>
> We emphasize that our work focuses on deepening the theoretical understanding of the generalization capabilities of world models—an area previously observed in empirical studies but lacking systematic theoretical analysis. Our experiments are specifically designed to support and validate the theoretical insights presented. For example, our "Robustness against Encoder Errors" experiment directly corroborates Corollary 3.8, which demonstrates how the model’s Jacobian norm can control the impact of errors on the loss $\mathcal{L}$.
>
> A4: _“The hyperparameter sensitivity for the regularization weight, $\lambda$, is expected to significantly affect the resulting model performance, but its sensitivity is not discussed (either theoretically or empirically) in the manuscript.”_
>
> We conducted experiments with \lambda values of 0.01, 0.05, and 0.1, and the results (in Appendix 6) indicate that the model's performance remained relatively stable across this range. Specifically, we did not observe significant variations in performance metrics, suggesting that our method is robust to moderate changes in \lambda. We are open to expanding our discussion of $\lambda$ sensitivity, including both theoretical considerations and additional empirical studies, in the future work.
>
> A5: _“There is a lack of comparison with existing approaches on addressing robustness and generalisability in world models.”_
>
> To the best of our knowledge, our work is the first to provide a theoretical understanding of the robustness and generalizability of world models. We discuss how our approach differs from existing research on robustness and generalizability in deep RL (Lines 124-133), specifically focusing on the unique challenges posed by latent representation errors.in world models.
>
> That said, if the reviewer could provide specific related theoretical work in world models on robustness and generalizability, we would be more than happy to include additional comparisons.
>
> [1] Soon Hoe Lim, N Benjamin Erichson, Liam Hodgkinson, and Michael W Mahoney. Noisy recurrent neural networks. Advances in Neural Information Processing Systems, 34:5124–5137, 2021.

---

### Official Review · Reviewer_p4R4 · 2024-11-04

**Soundness:** 2
**Presentation:** 2
**Contribution:** 3
**Rating:** 5
**Confidence:** 3

**Summary:**

This paper investigates the robustness and generalization of World Models by framing the learning process as a stochastic dynamical system within the latent state space. The authors present theoretical results to analyze the impact of latent representation errors on robustness and generalization in both zero-drift and non-zero-drift scenarios. Specifically, they demonstrate that, under certain assumptions, there exist CNN architectures for the world model’s encoder and decoder that achieve a small $W_1$ approximation error. Furthermore, they derive the relationship between the expected loss of the system in scenarios with no latent representation error and those with zero-drift and non-zero-drift errors. Based on their theoretical findings, the authors propose using Jacobian regularization to reduce representation error and improve the quality of rollouts during inference. Experiments were conducted on two environments, Walker and Quadruped, demonstrating that a world model trained with Jacobian regularization achieves faster convergence and greater robustness against unseen noisy observations, perturbed dynamics, and encoder errors.

**Strengths:**

- Overall, the paper is well-written and clear.

- This work enhances our understanding of the generalization and robustness of world models by leveraging less commonly used tools, specifically stochastic dynamical systems and differential geometry.

- The paper addresses a popular class of world models that have demonstrated practical effectiveness, adapting them to a theoretical framework with certain simplifying assumptions.

**Weaknesses:**

- The theorems are weakly interpreted in the discussion sections. Specifically, the primary variables and parameters within the theorems are not sufficiently explained, making it challenging for readers to fully understand the implications of the results. For instance, in Theorem 3.6, the authors suggest that $\varepsilon $ can be assumed to be small, but they do not provide details on the relevant parameters or the conditions under which this assumption is valid. Further interpretation and clarification of these variables and assumptions would significantly enhance the accessibility and impact of the theoretical results.

- The primary weakness of the paper lies in the experimental evaluation. The authors use DreamerV2 as the world model and compare results with and without Jacobian regularization. However, comparing only with the non-regularized version may be insufficient, as it lacks the same inductive bias present in the regularized model. I recommend that the authors include baselines that incorporate similar inductive biases to their technique—specifically, methods explicitly designed to be robust against perturbations at inference time.


- The experiments are limited to only two environments, Walker and Quadruped, making it difficult to fully assess the effectiveness and generalizability of the proposed method. I suggest that the authors expand their experiments to include additional environments.

**Questions:**

- Could the authors elaborate on how the results in table (1) connects with their theoretical findings?

- In (4), is the decoder intentionally conditioned on $\tilde{z}_t$? Dreamer and PlaNet typically condition the decoder on $z_t$ directly.

- Could the authors clarify what $\mathcal{X}$ represents in the CNN configuration? Is it intended to denote the state space or submanifold $\mathcal{M}$?

- Does Theorem 3.6 apply exclusively when the hidden representation is one-dimensional, given that the codomain of $f_{\text{CNN}}$ is $\mathbb{R}$?

- Could the authors elaborate on how "Theorem 4.1 correlates with the empirical findings in Hafner et al. (2019) regarding the diminished
predictive accuracy of latent states $\tilde{z}_t$ over the extended horizons"?

- Does "masked image percentage" refer to the portion of an image that has had noise applied?

- How did author select Jacobian regularization coefficient $\lambda$ in their experiments?

- In appendix D.6, what is the rationale behind using a different noise distribution for augmentation approach?

**Typos**

- Citations in the Related Work section are missing parentheses.

- Line 247: "choice choice" appears to be duplicated.

---

> ### Author Response · Authors · 2024-11-20
>
> Thank you for your feedback and questions!
>
> ### A1. Interpretation and Presentation of Theoretical Results
>
> - “The theorems are weakly interpreted in the discussion sections. Specifically, the primary variables and parameters within the theorems are not sufficiently explained, making it challenging for readers to fully understand the implications of the results.”
>
> We respectfully clarify that we have already provided detailed interpretations of the theoretical results throughout the paper. For instance:
>
> Theorem 3.6: “Theorem 3.6 indicates that with an appropriate CNN configuration, the W1 approximation error can be made to reside in a small region.. this allows us to apply the perturbation analysis of the dynamical system defined in Equations (5 - 8) in the following sections.” (Line 266 - 269)
>
> Theorem 3.7: “The presence of this Hessian-dependent term S, under latent representation error, implies a tendency towards wider minima in the loss landscape… ng… this insight resonates with the empirical results in Table 1 that model’s robustness gain is not significant when the error induced by large batch sizes is too small.” (Line 320 - 327)
>
> Corollary 3.8: “This observation also aligns with the theoretical insights in Lim et al. (2021) that the introduction of Brownian motion.. that could modulate the new bias term R˜ for stabilized implicit regularization.” (Line 355 - 367)
>
> Theorem 4.1: “Theorem 4.1 correlates with the empirical findings … the Jacobian norms within the latent dynamics model and the horizon length.” (406 - 409)
>
> Corollary 4.2: “ This corollary reveals that latent representation errors implicitly encourage exploration of unseen states … Intuitively, the stochasticity in the LDM also encourages greater exploration compared to its deterministic counterparts.” (Line 424 - 427)
>
> - “In Theorem 3.6, the authors suggest that it can be assumed to be small, but they do not provide details on the relevant parameters or the conditions under which this assumption is valid.”
>
> We respectfully note that Theorem 3.6 proves (instead of assume) that the W1 approximation error can indeed be confined to a small region, under an appropriate CNN configuration of the encoder-decoder architecture, rather than an arbitrary assumption of small error.
>
> We are happy to include further details to improve accessibility and are open to specific suggestions from the reviewer.
>
> ### A2. Experimental comparison to other robustness techniques
>
> We note that our comparison with data augmentation methods is detailed in Appendix D.6, due to space limitations in the main text. In a nutshell, our experiments show that models trained with Jacobian regularization outperform those using augmentation techniques, especially when confronted with perturbations not seen during augmentation. While state augmentation is effective for inference perturbations similar to those used in training, it would struggle to generalize to novel perturbations. In contrast, Jacobian regularization offers robustness that is less dependent on the diversity of augmented samples..
>
> ### A3.  On expanding experiments to more experiments
>
> We emphasize that our work focuses on deepening the theoretical understanding of the generalization capabilities of world models—an area previously observed in empirical studies but lacking systematic theoretical analysis. Our experiments are specifically designed to support and validate the theoretical insights presented so in this way experimental studies are complementary. For example, our "Robustness against Encoder Errors" experiment directly corroborates Corollary 3.8, which demonstrates how the model’s Jacobian norm can control the impact of errors on the loss $\mathcal{L}$.
>
> ### A4. Responses to Questions
>
> Q1: Connection of results in table (1) with main theoretical results
>
> Our somewhat surprising theoretical results  indicate that moderate latent representation errors can enhance the world model’s ability to handle perturbations, thereby improving both exploration and generalization. This behavior is analogous to the effect of gradient estimation errors observed during batch training, where a certain level of noise can act as implicit regularization.
>
> Specifically, we note that when the error $\bar{\sigma}_t(\cdot)$ is too small, the impact of the term S as a form of implicit regularization becomes less pronounced. Intuitively, this aligns with the empirical observations in Table 1, where the robustness gains are limited when errors induced by large batch sizes are insufficient to drive significant regularization effects. We further discuss this connection in Lines 324-327.
>
> Q2: “In (4), is the decoder intentionally conditioned on $\tilde{z}_t$?”
>
> The decoder is actually conditioned on $z_t$.Thank you for pointing out this typo.

---

> > ### Author Response · Authors · 2024-11-20
> >
> > Q3: “Could the authors clarify what $\mathcal{X}$ represents in the CNN configuration?”
> >
> > $\mathcal{X}$ denotes the domain over which the CNN function is defined. For instance, in the case of the encoder map, $\mathcal{X}$ is interpreted as $\mathcal{M}$, which the data distribution is supported on.
> >
> > Q4: “Does Theorem 3.6 apply exclusively when the hidden representation is one-dimensional, …?”
> > In Theorem 3.6, $f_\text{CNN}$ refers to maps from $\mathcal{M}$ to $\mathcal{Z}$ and from $\mathcal{Z}$ to $\mathcal{M}$, which correspond to the encoder and decoder maps, respectively. The special case involving a one-dimensional target $\mathcal{R}$ is only introduced in defining CNN configuration. We will clarify this distinction more explicitly in the revision.
> >
> > Q5: Insights on Theorem 4.1
> >
> > Theorem 3.6 highlights the diminished predictive accuracy of the latent states over extended horizons, which has also been observed empirically. Specifically, Theorem 4.1 indicates that the expected divergence arises from the accumulation of errors, depending on the magnitude of these errors.
> >
> > Q6: “Does "masked image percentage" refer to the portion of an image that has had noise applied?”
> >
> > Yes, that is correct.
> >
> > Q7: Jacobian regularization coefficients
> >
> > We experimented with $\lambda$ values of 0.01, 0.05, and 0.1. We did not observe significant performance variations across these choices of $\lambda$.
> >
> > Q8: Using a different noise distribution for augmentation approach
> >
> > We experimented with augmentations that included Gaussian noise and rotations. We would appreciate it if you could clarify what you mean by "using a different noise distribution for the augmentation approach" so that we can address your concern more specifically.

---

> ### Comment · Reviewer_p4R4 · 2024-11-21
>
> ## A1. Interpretation and Presentation of Theoretical Results
>
> > We respectfully clarify that we have already provided detailed interpretations of the theoretical results throughout the paper. For instance: ...
>
> Thank you for sharing examples of your interpretations. However, the reviewer has carefully reviewed these sections of the paper, which led to the use of the term "weak interpretation." The reviewer has also provided a specific example to clarify what is meant by "weak interpretation".
>
> > We respectfully note that Theorem 3.6 proves (instead of assume) that the W1 approximation error can indeed be confined to a small region, under an appropriate CNN configuration of the encoder-decoder architecture, rather than an arbitrary assumption of small error.
>
> It is recommended that the authors carefully revisit both their paper and the review comments to ensure clarity in their responses. The reviewer's wording aligns directly with the authors' explanation of Theorem 3.6, particularly the statement on line 268: _"In particular, this result indicates that the error magnitude ε in SDE (5) can be **assumed** to be small."_
>
> ## A2. Experimental comparison to other robustness techniques
> ## A3. On expanding experiments to more experiments
>
> > We note that our comparison with data augmentation methods is detailed in Appendix D.6, due to space limitations in the main text. In a nutshell, our experiments show that models trained with Jacobian regularization outperform those using augmentation techniques, especially when confronted with perturbations not seen during augmentation. While state augmentation is effective for inference perturbations similar to those used in training, it would struggle to generalize to novel perturbations. In contrast, Jacobian regularization offers robustness that is less dependent on the diversity of augmented samples..
>
> > We emphasize that our work focuses on deepening the theoretical understanding of the generalization capabilities of world models—an area previously observed in empirical studies but lacking systematic theoretical analysis. Our experiments are specifically designed to support and validate the theoretical insights presented so in this way experimental studies are complementary. For example, our "Robustness against Encoder Errors" experiment directly corroborates Corollary 3.8, which demonstrates how the model’s Jacobian norm can control the impact of errors on the loss .
>
>
> The paper claims to _propose a Jacobian regularization technique, supported by extensive experimental studies to demonstrate improvements in stability and robustness_. However, without additional comparisons to similar baselines and experiments across more environments, it becomes challenging to substantiate this claim. This is particularly critical given that two of the main contributions emphasize the practical benefits of Jacobian regularization. Furthermore, while Jacobian regularization is presented as a main contribution, the idea was previously introduced in another cited work.
>
> Thank you for further clarification on the questions!

---

> ### Author Response · Authors · 2024-11-22
>
> We greatly appreciate your prompt response to our rebuttal. We appreciate this further opportunity that allows us to improve the clarity of our work.
>
> A1. Interpretation and Presentation of Theoretical Results
>
> Thank you for pointing out the statements where our interpretation of theoretical results could be improved. To clarify, Theorem 3.6 proves (rather than assumes) that W1 approximation error can indeed be confined to a small region, given an appropriate CNN configuration for the encoder-decoder architecture. This result provides a justification for treating the perturbation as small as needed. In the revised version, we incorporate this clarification, particularly on Line 268 (now on Line 270-272, highlighted in red), to better connect the theorem to its implications for the subsequent perturbation analysis.
>
> A3. On expanding experiments to more experiments
>
> Thank you for your detailed observations regarding the positioning of Jacobian regularization in our work. We would like to clarify that the primary contribution emphasized in our paper is the theoretical understanding of the impacts of non-zero drift error on instability in world models, which shed light on how the well-known Jacobian regularization technique can be employed to mitigate this issue. We did not claim Jacobian regularization as a novel technique but rather as a theoretically grounded approach to address the challenges we have identified. To avoid any possible misinterpretation, we revised the draft to make this distinction more explicit in the introduction (on Line 102-104, Line 106-109, highlighted in red). We hope this has addressed your concern.
>
> Once again, we greatly appreciate your suggestions and feedback on our work. We are happy to address any further concerns.

---

### Author Response · Authors · 2024-11-20

At the outset, we find the “noise” in this review process very disturbing. We would like to point out that an earlier version was submitted to NeurIPS 2024 and received 7, 6, 6,  but was  rejected. The Area Chair’s primary critique was the lack of comparison with zero-shot deep RL methods, which is only tangentially related to our work given our focus on world models.

While we took the NeurIPS reject as bad luck, we were surprised and disheartened by the reviews  and feedback received for our ICLR 2025 submission, which show little appreciation of the theoretical work in filling the void to pursue a fundamental understanding of the generalization of world models, particularly given that the main critique is centered around calls for additional experimental studies—a comment that could be broadly applied to nearly any paper at ICLR (particularly those with theoretical focus). To the best of our knowledge, our work is the first to provide a theoretical understanding of the robustness and generalizability of world models. While we recognize that some degree of randomness is inevitable in peer review, the feedback we received in this case is really perplexing.

Please note that the main aim of our work is to obtain a theoretical understanding of the robustness and generalization capabilities of world models. This phenomenon, though observed in empirical studies, has not yet been systematically explored. Our paper specifically investigates the impact of latent representation errors on the world model’s generalization capability, addressing the error accumulation in latent dynamics — a challenge that is distinct from task transfer in policy-iteration based RL. It  develops an innovative SDE-based approach for understanding errors and generalization in learned latent dynamics models, and ultimately propose to regularize a learned dynamics model with the Jacobian w.r.t. the states. Experimental studies are complementary and are used to corroborate the theoretic finding.

---

### Note · Authors · 2024-12-03

I have read and agree with the venue's withdrawal policy on behalf of myself and my co-authors.